# Quantum variational algorithms are swamped with traps

Eric R. Anschuetz ●[1] ✉ & Bobak T. Kiani ●[2] ✉

One of the most important properties of classical neural networks is how surprisingly trainable they are, though their training algorithms typically rely on optimizing complicated, nonconvex loss functions. Previous results have shown that unlike the case in classical neural networks, variational quantum models are often not trainable. The most studied phenomenon is the onset of barren plateaus in the training landscape of these quantum models, typically when the models are very deep. This focus on barren plateaus has made the phenomenon almost synonymous with the trainability of quantum models. Here, we show that barren plateaus are only a part of the story. We prove that a wide class of variational quantum models—which are shallow, and exhibit no barren plateaus—have only a superpolynomially small fraction of local minima within any constant energy from the global minimum, rendering these models untrainable if no good initial guess of the optimal parameters is known. We also study the trainability of variational quantum algorithms from a statistical query framework, and show that noisy optimization of a wide variety of quantum models is impossible with a sub-exponential number of queries. Finally, we numerically confirm our results on a variety of problem instances. Though we exclude a wide variety of quantum algorithms here, we give reason for optimism for certain classes of variational algorithms and discuss potential ways forward in showing the practical utility of such algorithms.

The trainability of classical neural networks via simple gradient-based methods is one of the most important factors leading to their general success on a wide variety of problems. This is particularly exciting given the variety of no-go results via statistical learning theory, which demonstrates that in the worst case, these models are not trainable via stochastic gradient-based methods[1–4]. There has been recent hope that variational quantum algorithms (VQAs)—the quantum analog of traditional neural networks—may inherit these nice trainability properties from classical neural networks. Indeed, in certain regimes[5], training algorithms exist such that the resulting quantum model provably outperforms certain classical algorithms. This would potentially enable the use of quantum models to efficiently represent complex distributions which are provably inefficient to express using classical networks[6].

Unfortunately, such good training behavior is not always the case in quantum models. There have been previous untrainability results for deep VQAs due to vanishing gradients[7–10], and for nonlocal models due to poor local minima[11]; however, no such results were known for shallow, local quantum models with local cost functions. Indeed, there have been promising preliminary numerical experiments on the performance of VQAs in these regimes, but typically have relied on good initialization[12] or highly symmetric problem settings[13–15] to show convergence to a good approximation of the global optimum.

Here, we show that, generally, such models are not trainable, particularly when a good choice of initial point is not known and when the model does not exhibit a high amount of symmetry. We first prove general untrainability results in the presence of noise using techniques from statistical query learning theory. Surprisingly, these results hold

[1]MIT Center for Theoretical Physics, 77 Massachusetts Avenue, Cambridge, MA 02139, USA. [2]MIT Department of Electrical Engineering and Computer Science, 77 Massachusetts Avenue, Cambridge, MA 02139, USA. ✉e-mail: eans@mit.edu; bkiani@mit.edu

for all learning problems in a wide range of variational learning settings, and in many scenarios even when the magnitude of the noise is exponentially small in the problem size. We then consider the trainability of models that may not have noise by studying their typical loss landscapes. We prove that, for typical model instances, local minima concentrate far from the global optimum even for certain local shallow circuits that do not suffer from barren plateaus. This phenomenon can be visualized in Fig. 1, where the training landscape for a shallow QCNN learning a random instance of itself is shown to concentrate far from the global optimum. As in ref. 11, this phenomenon is the result of a trainability phase transition in the loss landscape of the quantum model. In ref. 11, this transition was governed by the ratio of the number of parameters to the Hilbert space dimension; we show in the shallow case that instead, this transition is governed by the ratio of the local number of parameters to the local Hilbert space dimension in the reverse light cone of a given measured observable. As this is typically much less than 1 for local variational ansatzes, these models are typically untrainable. We then give numerical evidence of this fact, and conclude by studying where there may be the reason for optimism in the training of certain variational quantum models.

## Results
### Preliminaries

Quantum machine learning algorithms have been a focus of intense research effort as potential use-cases for noisy, intermediate-scale quantum (NISQ)[16] devices. Just as in classical machine learning, algorithms are tasked with minimizing some risk:

$$\mathcal{R}(f) = \mathbb{E}_{\mathbf{x}}[\ell(f(\mathbf{x}))], \tag{1}$$

given a model $f$, a distribution of inputs $\mathbf{x}$, and a loss function $\ell$. To perform learning, one searches for a model $\widehat{f} \in \mathcal{F}$ in the function class $\mathcal{F}$ (e.g., the set of functions expressed by quantum neural networks). The expected risk $\mathcal{R}(\widehat{f})$ is typically not something one can calculate, as it requires access to the full probability distribution of the data. Instead, one often minimizes the empirical risk $\widehat{\mathcal{R}}(\widehat{f})$ (often named the training error) over a given training data set $D$:

$$\widehat{\mathcal{R}}(\widehat{f}) = \sum_{\mathbf{x}_i \in D} \ell(\widehat{f}(\mathbf{x}_i)). \tag{2}$$

Perhaps the most well-studied class of quantum machine learning algorithm consists of VQAs[17]. VQAs are a class of quantum generative models where one expresses the solution of some problem as the smallest eigenvalue and corresponding eigenvector (typically called the ground state) of an objective Hermitian matrix **H**−called the Hamiltonian−on $n$ qubits. Given a choice of generative model−often called an ansatz in the quantum algorithms literature:

$$|\boldsymbol{\theta}\rangle = \prod_{i=1}^{q} \mathbf{U_i}(\theta_i)|\psi_0\rangle \tag{3}$$

that for some choice of $\boldsymbol{\theta}$ is the ground state of **H**, the solution is encoded as the minimum of the loss function

$$\widehat{\mathcal{R}}_{\mathrm{VQE}}(\boldsymbol{\theta}) = \sum_{i=1}^{A} \alpha_i \langle \boldsymbol{\theta}|\mathbf{P}_i|\boldsymbol{\theta}\rangle, \tag{4}$$

where:

$$\mathbf{H} = \sum_{i=1}^{A} \alpha_i \mathbf{P}_i \tag{5}$$

is the Pauli decomposition of **H**. VQAs have found numerous applications[18], and a countless number of VQA instances have been proposed for various quantum learning tasks.

Typically, models in VQAs come in one of two flavors: Hamiltonian agnostic models, and Hamiltonian informed models. Hamiltonian agnostic models are constructed such that the $\mathbf{U_i}$ are independent of **H**, and are generally chosen to be efficient to implement. This is most analogous to the case in classical generative modeling, where the model structure is usually independent from the specific choice of data **H**. One might hope then that training Hamiltonian agnostic VQAs is completely analogous to the classical setting, then, and the loss landscape of (4) exhibits the desirable properties that enable trainability found in classical networks[19,20].

Unfortunately, unlike the classical setting, the performance of VQAs is often dominated by poor performance in the training procedure (see Supplementary Note 1 for a discussion). For one, VQAs tend to exhibit barren plateaus when they are deep; namely, gradients of deep variational quantum circuits vanish exponentially with the problem size in many settings[7,8,10]. Problematic training in this regime has also been studied beyond gradient descent[21,22].

Until recently, less was known about the trainability of VQAs in the shallow model regime. Numerically, refs. 13, 23 showed that randomly chosen variational landscapes typically have poor local minima, a result which was later proven in ref. 11 for nonlocal models using tools from random matrix theory. In a similar line of research, ref. 24 showed

(a)

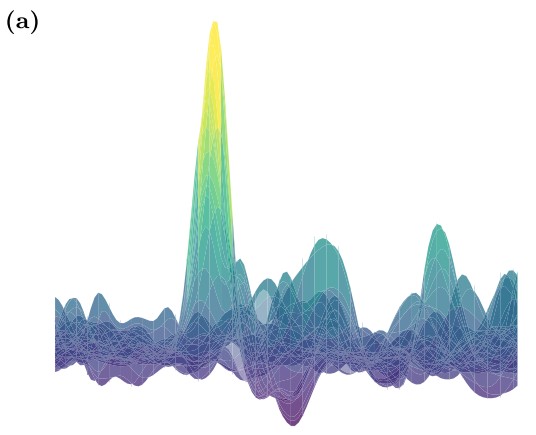

(b)

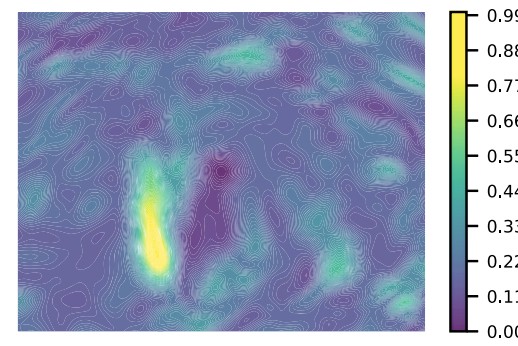

**Fig. 1 | Typical shape of loss landscape.** Loss landscapes of underparameterized quantum variational algorithms generally appear "bumpy," filled with various local minima and traps. Here, we plot the loss landscape as a **a** surface and **b** contour plot along two random normalized directions for the teacher-student learning task of the QCNN for 14 qubits. Though a global minimum is located at the center of the plot, finding this global minima is generally challenging due to the shape of the loss landscape. Details of this visualization are given in Supplementary Note 6.

**Table 1 | A summary of previous results on the untrainability of variational quantum algorithms**

| Result | Dimension | Locality | Depth | Worst case? | Barren plateaus? | Poor minima? |
|---|---|---|---|---|---|---|
| Ref. 7 | $d$ | 2 | $\Omega(n^{\frac{1}{d}})$ | ✗ | ✓ | ? |
| Ref. 8 | 1 | 2 | $\omega(\log(n))$ | ✗ | ✓ | ? |
| Ref. 10 | $d$ | 2 | $\omega(\log(n)^{\frac{1}{d}})$ | ✗ | ✓ | ? |
| Ref. 11 | N/A | $n$ | $\Omega(1)$ | ✗ | ✓/✗ | ✓/✗ |
| Ref. 24 | $d$ | 2 | $\Omega(1)$ | ✓ | ? | ✓ |
| Our results | $d$ | 2 | $\Omega(1)$ | ✗ | ✗ | ✓ |

A label of "✓/✗" denotes that the paper studied certain regimes where the phenomenon was present, and certain regimes where it was not. A label of "?" denotes that the phenomenon was not studied. "Dimension" indicates the locality structure of the ansatzes study. For instance, Dimension = 1 denotes ansatzes with nearest-neighbor interactions for qubits on a line. "Worst case" denotes analysis performed with adversarial data.

**Table 2 | Relatively simple classes of functions require exponentially many statistical queries to learn using any naive algorithm that reduces to statistical queries**

| Setting ($n$ qubits, $L$ layers) | Query complexity ($\beta < 1/2$[a]) |
|---|---|
| $L = 1$, global measurement, single qubit gates | $2^{\Omega(n)}$ if $\tau \geq 3^{-\beta n}$ |
| $L = \lceil \log_2 n \rceil$, single qubit measurement, global 1- and 2-local gates | $2^{\Omega(n)}$ if $\tau \geq 4^{-\beta n}$ |
| $L \ll n$, single qubit measurement, neighboring 1- and 2-local gates on a $d$-dim. lattice | $2^{\Omega(L^d)}$ if $\tau = \Omega(1)$[b] |
| $L = 1$, single qubit gates, unitary learning | $2^{\Omega(n)}$ if $\tau \geq 4^{-\beta n}$ |

[a]Technically, we require $\beta = 1/2 - \Omega(1)$.
[b]$\tau = 2^{-\omega\left(\min\left(2L, n^{1/d}\right)^d\right)}$ is sufficient.
The table above quantifies the number of queries needed to identify a target function in the function class, over a distribution of states that forms a 2-design and with queries that have tolerance $C_{max}\tau$ (query tolerance lower bounded by a constant times $C_{max}$ suffices in all cases).

that for certain quantum variational ansatzes or quantum neural networks, there exist data sets and loss functions which induce exponentially many local minima in the loss landscape. Refs. 25, 26 both showed that, in an overparameterized regime, these models experience good local minima, though this transition to trainability typically occurs at an intractable number of parameters. Finally, assuming the presence of a constant rate of noise per ansatz gate, ref. 27 showed convergence of the loss landscape to the uniform distribution at a rate exponential in the circuit depth. Many of these previous results on the untrainability of VQAs are summarized in Table 1, along with a summary of our results which focus on the shallow, local regime.

**Learning in the statistical query framework**

Quantum machine learning algorithms are inherently noisy due to both unavoidable sources of error—such as shot noise from sampling outputs—or potentially correctable sources of error, such as gate errors and state preparation noise. In such noisy settings, the statistical query (SQ) model provides a useful framework for quantifying the complexity of learning a class of functions by considering how many query calls to a noisy oracle are needed to learn any function in that class (see Supplementary Note 2 for a brief review and history of SQ models)[2,28,29]. In this setting, we consider the optimization of a risk of the form of (2). We assume there is a target observable $\mathbf{M}$ that we would like to learn on some distribution over states $\mathcal{D}$. We define a correlational statistical query $qCSQ(\mathbf{O}, \tau)$, which takes in a bounded observable $\mathbf{O}$ with $\|\mathbf{O}\| \leq 1$ and a tolerance $\tau$ and returns a value in the range:

$$\mathbb{E}_{\rho \sim \mathcal{D}}[Tr(\mathbf{O}\rho)Tr(\mathbf{M}\rho) - \tau] \leq qCSQ(\mathbf{O},\tau) \leq \mathbb{E}_{\rho \sim \mathcal{D}}[Tr(\mathbf{O}\rho)Tr(\mathbf{M}\rho) + \tau].$$

$$(6)$$

Note that there are no guarantees on the form of the additive error other than it is within the tolerance $\tau$, and may, for instance, depend on the observable being queried $\mathbf{O}$. Though SQ oracle calls

may at first appear unrelated to variational algorithms, we show in the Methods that many common variational optimizers in the presence of noise of the magnitude $\tau$ reduce to calls to an SQ oracle; for instance, commonly used first and second order optimization algorithms fall within the framework of the SQ model we consider. In the Methods, we also describe an analogous SQ model for learning unitaries.

To quantify the hardness of learning variational circuits, we consider the task of learning certain function classes generated by shallow variational circuits over a distribution of inputs $\mathcal{D}$ which forms a 2-design. Our results also generally hold for distributions that are uniform over states in the computational basis, recovering the statistical query setting for classical Boolean functions. Table 2 summarizes the number of queries needed to learn various function classes which are generated by variational circuits, with proofs deferred to Supplementary Note 3. In all settings we consider, an exponential number of queries (in either $n$ or the light cone size) are needed to learn simple classes, such as the class of functions generated by single qubit gates followed by a fixed global measurement. This hardness intuitively arises because each individual query can only obtain information about a few of the exponentially many orthogonal elements in the function class. More formally, we lower bound the SQ dimension (defined in the "Methods") of the function classes considered in Table 2 to show our query lower bounds.

Our hardness results hold for any target observable $\mathbf{M}$, as long as the learning setting is one we consider in Table 2. Furthermore, they hold for any variational ansatz—not just on average—provided it is in one of the settings of Table 2. Finally, our results hold for any constant error $\tau$ in the statistical queries; indeed, the majority of our results hold even if this noise were only exponentially small in the problem size. For instance, training via gradient descent where the gradient is estimated using polynomially many samples fits into this framework immediately just from the induced shot noise.

In a more positive light, learning local Hamiltonians generated by shallow depth circuits can potentially be efficiently performed as the complexity grows exponentially only with locality or depth in this setting. In fact, prior results have provably shown that certain classes of Hamiltonians are efficiently learnable using properly chosen algorithms[30,31]. Nevertheless, this does not correspond to efficient learnability of the ground state of a given Hamiltonian, as learnability of the properties of a Hamiltonian is not the same as the learnability of its ground state. Indeed, we will see in section 'Loss landscapes of local variational quantum algorithms' that typically, even in this setting, learning the ground state of such a local Hamiltonian is difficult.

Our hardness results do not indicate that simple classes of functions like those generated by single qubit rotations are hard to learn for all algorithms, but only those whose steps reduce to statistical queries. For example, the class of Pauli channels is not learnable in the SQ setting, but there exist simple, carefully constructed, algorithms which can learn Pauli channels[32–34]. This is analogous to the classical setting where parity functions are hard to learn in the noisy SQ setting, but efficient to learn using simple linear regression[29]. Similarly, the related

work of ref. 35 showed that output distributions of Clifford circuits can be hard to learn using statistical queries, but efficient using a technique that resorts to linear regression on a matrix formed from samples of the overall distribution. More loosely, our results provide support to the basic maxim that algorithms which apply too broadly will work very rarely[36]; more careful construction of learning algorithms tailored to the problem at hand is generally necessary. One straightforward way to avoid the hardness of the SQ setting is to construct algorithms whose basic steps do not reduce to statistical queries, e.g., via the construction of non-global metrics[37–39]. However, such a fix is by no means guaranteed to avoid the more general issues of poor landscapes and noise that also make learning in the SQ setting so difficult, as we now examine.

### Loss landscapes of local variational quantum algorithms

We now consider the trainability of VQAs in the noise-free regime, beyond optimization algorithms that reduce to statistical queries. Though we are unable to prove the very strong no-go results proved in the SQ framework, we are able to show that the loss landscapes of typical local variational algorithms with Hamiltonian agnostic ansatzes are unamenable to optimization. We achieve this by showing that typically, the loss landscapes of shallow, local VQAs are swamped with poor local minima.

As discussed in Table 1, it is already known that deep Hamiltonian agnostic ansatzes are typically untrainable due to the presence of barren plateaus[7,8,10]; hence, here, we focus on shallow ansatzes. Previous results[11] have also shown that shallow, nonlocal models are untrainable, by showing that the scrambling of variational ansatzes over the entire system in these instances induce poor local minima. These techniques were not extendable to shallow, local ansatzes, however, which do not scramble globally.

Instead, here, we show that ansatzes that approximately scramble locally are difficult to train. As we will later show, this includes common classes of variational ansatzes, such as Hamiltonian agnostic checkerboard ansatzes on a $d$-dimensional lattice. We show that this approximate, local scrambling suffices to imply that the loss landscapes of these VQAs are close to those of Wishart hypertoroidal random fields (WHRFs). These are random fields parameterized by $l, m$ of the form:

$$F_{WHRF}(\mathbf{w}) = m^{-1} \sum_{i,j=1}^{2^l} w_i J_{i,j} w_j, \tag{7}$$

where $\mathbf{J}$ is drawn from a Wishart distribution with $m$ degrees of freedom, and $\mathbf{w}$ are points on a certain embedding of the hypertorus $(S^1)^{\times l}$ in $\mathbb{R}^{2^l}$. We demonstrate this convergence via new techniques, directly bounding the error in the joint characteristic function of the function value, gradient, and Hessian components of the variational loss from those of WHRFs. As the typical loss landscapes of WHRFs are known given these random variables (see "Methods" for a summary), by demonstrating sufficient convergence of these random variables to those of WHRFs, we will be able to infer the distribution of critical points for local VQAs.

To begin, we take our (assumed traceless) problem Hamiltonian to have Pauli decomposition:

$$\mathbf{H} = \sum_{i=1}^{A} \alpha_i \mathbf{P_i}, \tag{8}$$

and for simplicity scale and shift the loss landscape of (4) to be of the form:

$$\widehat{\mathcal{R}}_{VQE}(\boldsymbol{\theta}) = 1 + \| \boldsymbol{\alpha} \|_1^{-1} \sum_{i=1}^{A} \alpha_i \langle \boldsymbol{\theta} | \mathbf{P_i} | \boldsymbol{\theta} \rangle, \tag{9}$$

where $\boldsymbol{\alpha}$ is the vector of all $\alpha_i$ and the ansatz $|\boldsymbol{\theta}\rangle$ is as given in (3). As this ansatz is assumed to be shallow and local, we assume that the reverse light cone of each $\mathbf{P_i}$ under the ansatz is of size $l \ll n$.

As in most analytic treatments of Hamiltonian agnostic VQAs, we consider certain randomized classes of ansatzes[7,8,11]. Roughly, we assume that in a local region around each measured Pauli observable $\mathbf{P_i}$ the ansatz is an $\epsilon$-approximate $t$-design; that is, its first $t$ moments are $\epsilon$-close to those of the Haar distribution. This is a much weaker assumption than global scrambling of the ansatz. For instance, for $\mathbf{P_i}$ of constant weight, such approximately locally scrambling circuits include constant depth local circuits with random local gates[40]. We discuss in more detail when this assumption holds in practice when specializing to common variational quantum learning scenarios, and defer technical details to Supplementary Note 4.

Our main result, informally, is that the random field given by (9) under this approximate, local scrambling assumption converges in distribution to that of a WHRF. The formal statement and derivation of this result are given in Supplementary Note 4, where we also lay out our assumptions more explicitly. Informally, the result follows by deriving a bound on the error in the joint characteristic function of the loss function and its first two derivatives from that of a WHRF. We then use this to bound the error in distribution that is incurred by the induced scrambling being only approximate. Finally, we show using properties of local Haar random gates and the locality of the problem Hamiltonian that this suffices to prove the convergence of these random variables to those of a WHRF.

**Theorem 1.** (Approximately locally scrambled variational loss functions converge to WHRFs, informal). Let

$$m \equiv \frac{\| \boldsymbol{\alpha} \|_1^2}{\| \boldsymbol{\alpha} \|_2^2} 2^{l-1} \tag{10}$$

be the degrees of freedom parameter. Assume $q \log(q) = o(m)$, where $q$ is the number of ansatz parameters in the reverse light cone of each $\mathbf{P_i}$. Then, the distribution of (9) and its first two derivatives are equal to those of a WHRF

$$F_{WHRF}(\boldsymbol{\theta}) = m^{-1} \sum_{i,j=1}^{2^l} w_i J_{i,j} w_j \tag{11}$$

with $m$ degrees of freedom, up to an error in distribution on the order of $\tilde{O}(poly(\frac{1}{t} + \epsilon + \exp(-l)))$. Here, $\mathbf{w}$ are points on the hypertorus $(S^1)^{\times l}$ parameterized by $\tilde{\boldsymbol{\theta}}$, where $\tilde{\theta}_i$ is the sum of all $\theta_j$ on qubit $i$.

We interpret this result as the degrees of freedom $m$ of the model being given by roughly the sum of the local Hilbert space dimensions of the reverse light cones of terms in the Pauli decomposition of $\mathbf{H}$. We interpret this as the local underparameterization of the model, to be contrasted with the global underparameterization interpretation when $m$ is exponentially large in $n$. Using known properties of the loss landscapes of WHRFs (see "Methods"), we are then able to prove the following result on the loss landscapes of local VQAs:

**Corollary 2.** (Shallow, local VQAs have poor loss landscapes, informal). Let $\widehat{\mathcal{R}}_{VQE}$ be a local VQA loss function of the form of (9). Assume all coefficients $\alpha_i$ of the Pauli decomposition of $\mathbf{H}$ are $\Theta(1)$, and

$$l \log(n) + q \log(q) = o\left(2^l A\right) \tag{12}$$

Then $\widehat{\mathcal{R}}_{VQE}$ has a fraction superpolynomially small in $n$ of local minima within any constant additive error of the ground state energy.

Optimizing loss landscapes where only a superpolynomially small (in $n$) fraction of the local minima are near the global minimum in energy is expected to be difficult. Indeed, algorithms such as gradient

descent would then expect to have to be restarted a superpolynomial number of times before finding a good approximation to the global minimum; we also give heuristic reasons why this should continue to be true for other local optimizers in Supplementary Note 7. Our results stand in stark contrast with the loss landscapes typically found in classical machine learning, where almost all local minima closely approximate the global minimum in function value[19,20].

In the shallow ansatz regime—where $q, l = O(\text{polylog}(n))$—and assuming an extensive problem Hamiltonian such that $A = \Omega(n)$, the condition given by (12) is always satisfied. Interestingly, this is a regime where barren plateaus are known not to occur[10], demonstrating that poor local minima can give rise to poor optimization performance even when the loss function features large gradients. We now specialize to common variational quantum learning scenarios, and consider the implications of Corollary 2.

First, let us consider $d$-dimensional checkerboard ansatzes of constant depth. Fix $p, t$ to be sufficiently large constants. We assume that the initial state forms an $O\left(\frac{1}{\text{poly}(n)}\right)$-approximate $t$-design on $l$ qubits around each Pauli observable of weight $k$; this can be done via a depth $p$, $d$-dimensional circuit of 2-local Haar random unitaries when $l = O\left(\frac{(p+k)^d}{\text{poly}(t)}\right) \geq k$ for some fixed polynomial in $t$[40,41]. After this state preparation circuit, a traditional depth $\Theta\left(l^{\frac{1}{d}}\right)$ (i.e., independent of $n$), $d$-dimensional, $n$-qubit checkerboard circuit is applied, with observable reverse light cones of size at greatest $l$. By Corollary 2, these variational ansatzes are untrainable due to poor local minima, yet by the results of ref. 10 do not suffer from barren plateaus.

One interesting consideration is extending this result to traditional checkerboard ansatzes, without the special state preparation procedure we have considered. There, the $l = O\left(\frac{(p+k)^d}{\text{poly}(t)}\right)$ qubit local state is mixed, and our results, therefore, do not directly apply. However, we expect no reason for the mixedness of the initial state to improve training performance in any way. We validate this intuition numerically in section 'Numerical results'.

We also consider a class of models similar to quantum convolutional neural networks (QCNNs)[12] previously shown not to suffer from barren plateaus[42]. Though these models are in full generality trained on arbitrary loss functions, for learning various physical models the loss may take the form of (4). QCNNs are defined by their measurement of a subset of qubits at periodic intervals, via so-called pooling layers; for sufficiently deep (i.e., large constant depth) convolutional layers, then, at some point in the model, the number of remaining qubits will be sufficiently small such that the remaining convolutional layers are approximately scrambling. If one then assumes that the initial states are adversarially chosen such that they remain pure by this layer, this scenario reduces to the shallow checkerboard ansatz scenario, and once again we expect poor local minima by Corollary 2. Even if the initial states are not adversarially chosen and the input to the scrambling convolutional layers is mixed, we expect by similar intuition the model to remain untrainable; we will see this numerically, where we also observe that this poor training occurs when training on loss functions beyond (4).

## Numerical results

To numerically validate our theoretical findings, we perform numerical simulations showing that learning in various settings cannot be guaranteed unless exponentially many parameters are included in an ansatz. We only consider problems and ansatzes where the existence of a zero loss global minimum is guaranteed to study whether or not optimizers can actually find the global minimum or a similarly good critical point. We parameterize all trainable 2-qubit gates in the Lie algebra of the four-dimensional unitary group, and implement the resulting unitary matrix via the exponential map which is surjective and capable of expressing any local $4 \times 4$ unitary gate. In all cases, we perform simulations using calculations with computer precision and analytic forms of the gradient (see Supplementary Note 6 for more

details). In practice, actual quantum implementations will be hampered by various sources of inefficiency, such as the lack of an analogous method of backpropagation for calculating gradients, sampling noise, or even gate errors. Thus, our numerical analysis can be interpreted as a "best case" setting for quantum computation where we disregard such inefficiencies and focus solely on learnability. In Supplementary Note 5, we further study variations of the teacher-student learning and random variational quantum eigensolver (VQE)[17] settings discussed here. We also consider the training performance of VQE in finding the ground state of a Heisenberg XYZ Hamiltonian[43]. Our supplemental results reinforce our findings here.

One may conjecture that it is plausible to learn the class of functions generated by relatively shallow depth variational teacher circuits by parameterizing a shallow-depth student circuit of the same form and training its parameters. In this so-called teacher-student setup, we are guaranteed the existence of a perfect global minimum since recovering the parameters of the teacher circuit achieves zero loss. In other words, the global minimum is guaranteed to be achievable in the setting we consider here. Still, we showed earlier that such circuits typically have many poor local minima, and are always hard to learn in the statistical query setting. Here, we provide numerical evidence of these findings for the QCNN ansatz. Additional confirmation of these findings with a checkerboard ansatz is included in Supplementary Note 5.

The QCNN presents an interesting test bed for our analysis since it has been shown in prior work to avoid barren plateaus[42]. Nevertheless, the QCNN, like other models, is riddled with poor local minima in generic learning tasks. For our analysis, we attempt to learn randomly generated QCNNs with a parameterized QCNN of the same form. In the QCNN, both student and teacher circuits have parameterized 2-qubit gates at each layer followed by 2-qubit pooling layers (see Supplementary Note 6 for more details). Each 2-qubit gate is fully parameterized in the Lie algebra of the unitary group. Networks are trained to predict the probability of the measurement of the last qubit in the teacher circuit. In other words, the student network is trained on a classification problem defined by the teacher network where, by construction, perfect classification accuracy is known to be achievable. We benchmark performance with the classification accuracy, where a prediction is considered correct when it predicts the most likely measurement of the last qubit correctly. Networks are trained via the Adam optimizer[44] to learn outputs of 512 randomly chosen computational basis states. QCNNs with 4, 8, 12, and 16 qubits have 32, 48, 64, and 64 trainable parameters, respectively.

Figure 2 plots the final training accuracy achieved over 100 random simulations for varying ranges of circuit sizes. For circuits with 4 qubits, the training is sometimes successful, often achieving an accuracy above 85 percent on the training dataset. However, as the number of qubits grows, even past 8 qubits, the optimizer is unable to recover parameters which match the outputs of the teacher circuit. The results here show that the QCNN circuit—which has $O(\log n)$ depth—still scrambles outputs to hinder learnability.

We now consider VQE. To analyze the performance of variational optimizers, we consider problems and ansatzes which are capable of recovering the global minimum. We aim to find the ground states of local Hamiltonians $\mathbf{H}_t$ over $n$ qubits that take the form of single qubit Pauli $\mathbf{Z}$ Hamiltonians conjugated by $L^*$ layers of two alternating unitary operators $\mathbf{U}_1$ and $\mathbf{U}_2$, which are product unitaries on neighboring 2-local qubits:

$$\mathbf{H}_t = \left(\mathbf{U}_2^\dagger \mathbf{U}_1^\dagger\right)^{L^*} \left[\sum_{i=1}^{n} \mathbf{Z}_i\right] \left(\mathbf{U}_1 \mathbf{U}_2\right)^{L^*} + n\mathbf{I}. \tag{13}$$

The added identity matrix normalizes the Hamiltonian to have ground state with energy 0. Since the ground state of $\sum_{i=1}^{n} \mathbf{Z}_i$ is the state $|1\rangle^{\otimes n}$, we are guaranteed the existence of a global minima when

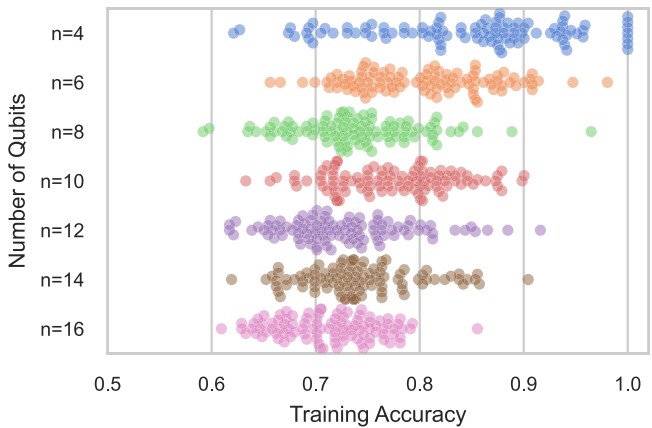

**Fig. 2 | Teacher-student evaluation for the *n*-qubit QCNN.** The student circuit is unable to learn the teacher circuit as the number of qubits grows, converging to a local minimum of the loss landscape. The existence of a global optimum is guaranteed as the teacher circuit is drawn from a random initialization of the same QCNN structure of the student circuit. Here, for a ranging number of qubits, 100 student circuits are trained to learn randomized teacher circuits of the same form and the resulting swarm plots of the final training accuracy are shown.

using a checkerboard ansatz of at least depth $L^*$, since this ansatz can "undo" the conjugation by unitary operators. In the remainder of this section, we consider (13) with $L^* = 4$.

We measure the performance of optimization with two metrics. The first is the loss function itself, which is the average energy $\langle \psi | \mathbf{H}_t | \psi \rangle$ of the VQE ansatz state $|\psi\rangle$ for the given Hamiltonian $\mathbf{H}_t$. The second is the trace distance to the ground state $|\phi_g\rangle$ of $\mathbf{H}_t$, equal to $\| |\phi_g\rangle\langle\phi_g| - |\psi\rangle\langle\psi| \|_1 / 2$. Both of these metrics converge to zero at the global minimum.

We first aim to learn the ground state using a checkerboard ansatz by performing vanilla gradient descent on $L = L^* = 4$ parameterized layers, equal in depth to the Hamiltonian conjugation circuit and thus capable of recovering the ground state. In Fig. 3a, we plot the final values of the loss and trace distance for 24 randomly initialized VQE problems for a number of qubits ranging from 4 to 24. Similar results are observed when using more advanced optimizers such as Adam (see Supplementary Note 6)[44]. Consistent with our theoretical findings, convergence clusters around local minima far from the ground state, particularly as the number of qubits grows.

Our theoretical results also imply the difficulty of training beyond a finite fraction of the ground state energy in a VQE setting. Figure 3b illustrates this phenomenon when performing optimization on a 14 qubit ansatz. As more parameters are added to the ansatz via increasing its depth $L$, the VQE algorithm performs better, but it is not until the number of parameters is exponential in the problem size that convergence to a global minimum (or even within a small additive error of the global minimum) is guaranteed. This is true even though the ansatz is capable of expressing the ground state at $L = 4$. Simulations here are performed as before on random $L^* = 4$ Hamiltonians of the form of (13).

## Discussion

Though VQAs—and quantum machine learning models in general— have been cited as perhaps the most promising use case for quantum devices in the near future[16], theoretical guarantees of their training performance have been sparse. Here, we have excluded a wide class of variational algorithms by showing that in many settings, they are in fact not trainable. We showed this in two different frameworks: first, in section 'Learning in the statistical query framework', we studied various classes of quantum models in the statistical query framework. We showed that in the presence of noise, exponentially many queries in

the problem size are needed for these models to learn. As a complementary approach, we also examined the typical loss landscapes of VQAs in the noiseless setting in section 'Loss landscapes of local variational quantum algorithms', and showed that even at constant depth these models can have a number of poor local minima superpolynomially large in the problem size. We also numerically confirmed these results for a variety of problems in section 'Numerical results'. These results go beyond the typical studies on the presence of barren plateaus, as many of the models we study here have gradients vanishing only polynomially quickly in the problem size. Our work demonstrates that showing that barren plateaus are not present in a model does not necessarily vindicate it as trainable.

These results, though they exclude a wide variety of VQAs, still leave room for hope in the usefulness of these algorithms. Particularly, our analysis in the noiseless setting of landscapes of VQAs focuses on very general, Hamiltonian agnostic ansatzes; in various instances, more focused ansatzes may be trainable. For instance, as previously shown in ref. 5, for certain classes of problems the quantum approximate optimization algorithm (QAOA)[45] is provably able to outperform the best unconditionally proven classical algorithms, even when taking into account the training of the model. This is due to parameter concentration, where the global optimum for small problem instances is close to the global optimum for large problem instances[46]. These results demonstrate the power of good model initialization in VQAs: even if the total variational landscape is swamped with poor local minima, good initialization may ensure that the optimizer begins in the region of attraction of the global minimum. Though this is perhaps most relevant for the VQE[17] and QAOA[45] where there exists physical intuition for potentially performant parameter initializations, in more traditional machine learning settings this may manifest as good performance on certain inputs to the model.

Variationally studying models with many symmetries may also avoid our poor performance guarantees. Intuitively, our results here are the consequence of underparameterization. Namely, unless the ansatz is parameterized such that the number of parameters grows with the (local) Hilbert space dimension, the model is not trainable. Typically, this Hilbert space dimension is exponentially larger than the number of parameters the ansatz uses to explore it. However, if the model is heavily constrained by symmetries, this dimension might be much smaller. Such models were studied numerically in refs. 13, 47, where it was shown that certain VQAs optimize efficiently. Though often these models can be solved classically when the symmetries are known, these symmetries may not be known a priori. Indeed, one may be able to test for the presence of symmetries in a given model by studying whether associated VQAs are trainable. Similar to these general symmetry considerations, known structure in the problem may also allow one to build up hierarchical ansatzes that are able to be trained sequentially. We leave further investigation in these directions to future work.

Finally, though many variational models fit the framework of (4), there exist other settings of VQAs. One class of such models includes quantum Boltzmann machines, which attempt to model given quantum states via the training of quantum Gibbs states[48]. When the full quantum Gibbs state is observed, it is known that these models are efficiently trainable[30], and numerically it is known that these models are trainable even when the full state is not observed[48,49]. Furthermore, though in full generality preparing quantum Gibbs states is difficult, state preparation has been shown to be efficient in certain regimes relevant to machine learning[49–51], potentially giving an end-to-end trainable quantum machine learning model. We leave further analytical investigation on the training landscapes of quantum Boltzmann machines to future work.

Our results contribute to the already vast library of literature on the trainability of variational quantum models in further culling the landscape of potentially trainable quantum models. We hope these

(a)
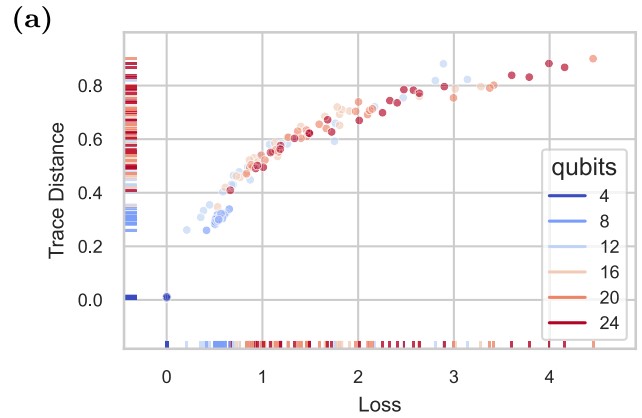

(b)
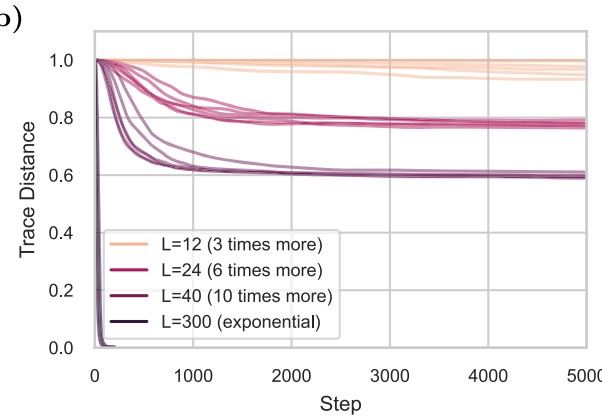

**Fig. 3 | Empirical analysis of VQE. a** Scatter plot of the final loss and trace distance of the VQE state after 30,000 steps of gradient descent optimization shows that the algorithm converges to poorer local minima as the number of qubits grows. 24 simulations are performed for each value of *n*. The algorithm always succeeds at obtaining the ground state with 4 qubits, but progressively struggles more with added qubits. **b** The number of layers needed to guarantee convergence to the ground state empirically grows exponentially with the number of qubits. Here, we consider 4-layer Hamiltonians of the form of (13) on 14 qubits where the number of layers *L* in the ansatz is varied. When the ansatz has 300 layers—enough that the number of ansatz parameters is larger than the explored Hilbert space dimension—the model successfully converges to the ground state, rather than remaining stuck in a poor local minimum.

---

results have the effect of focusing research efforts toward classes of models that have the potential for trainability, and whittle down the search for practical use cases of VQAs.

## Methods

### The statistical query learning framework

We give a brief overview of the classical SQ model here, and provide a more detailed review in Supplementary Note 2. Given an input and output space $\mathcal{X}$ and $\mathcal{Y}$, let $\mathcal{D}$ be a joint distribution on $\mathcal{X} \times \mathcal{Y}$. In the classical SQ model, one queries the SQ model by inputting a function $f$ and receiving an estimate of $\mathbb{E}_{(x,y) \sim \mathcal{D}}[f(x,y)]$ within a given tolerance $\tau$. As an example, one can query a loss function $\ell$ for a model $m_{\boldsymbol{\theta}}$ with parameters $\boldsymbol{\theta}$ by querying the function $\ell(m_{\boldsymbol{\theta}}(x), y)$. A special class of statistical queries are inner product queries where query functions $g$ are defined only on $\mathcal{X}$ and the correlational statistical query returns an estimate of $\mathbb{E}_{(x,y) \sim \mathcal{D}}[g(x) \cdot y]$ within a specified tolerance $\tau$.

In detail, the SQ models we consider take the forms below:

**Definition 3.** (Quantum correlational statistical query (qCSQ)). Assume there is a target observable **M** that we would like to learn on some distribution over states $\mathcal{D}$. Applying the correlational SQ model to the quantum setting, we define the query $qCSQ(\mathbf{O}, \tau)$ which takes in a bounded observable **O** with $\|\mathbf{O}\| \le 1$ and a tolerance $\tau$ and returns a value in the range:

$$\mathbb{E}_{\rho \sim \mathcal{D}}[Tr(\mathbf{O}\rho)Tr(\mathbf{M}\rho) - \tau] \le qCSQ(\mathbf{O}, \tau) \le \mathbb{E}_{\rho \sim \mathcal{D}}[Tr(\mathbf{O}\rho)Tr(\mathbf{M}\rho) + \tau].$$

(14)

**Definition 4.** (Quantum unitary statistical query (qUSQ)). In the unitary compilation setting, one aims to learn a target unitary transformation $\mathbf{U}_*$ over a distribution $\mathcal{D}$ of input/output pairs of that unitary transformation. Here, the oracle $qUSQ(\mathbf{V}, \tau)$ takes in a unitary matrix **V** and a tolerance $\tau$ and returns a value in the range:

$$\mathbb{E}_{\rho \sim \mathcal{D}}\left[Re[Tr(\mathbf{U}_*^\dagger \mathbf{V}\rho)] - \tau\right] \le qUSQ(\mathbf{V}, \tau) \le \mathbb{E}_{\rho \sim \mathcal{D}}\left[Re[Tr(\mathbf{U}_*^\dagger \mathbf{V}\rho)] + \tau\right].$$

(15)

Importantly, if $\mathcal{D}$ is a 1-design over $n$ qubit states, then the above can be simplified using the formula $\mathbb{E}_{\rho \sim \mathcal{D}}[Re[Tr(\mathbf{U}_*^\dagger \mathbf{V}\rho)]] = 2^{-n}Re[Tr(\mathbf{U}_*^\dagger \mathbf{V})]$ (see proof in Supplementary Note 3). Queries to *qUSQ* are related to

performing a Hadamard test[52], also a common subroutine in variational algorithms[53].

The queries above take the forms of inner products, with $\langle \mathbf{M}_1, \mathbf{M}_2 \rangle_{\mathcal{D}} = \mathbb{E}_{\rho \sim \mathcal{D}}[Tr(\mathbf{M}_1\rho)Tr(\mathbf{M}_2\rho)]$ and $\langle \mathbf{U}_1, \mathbf{U}_2 \rangle_{\mathcal{D}} = \mathbb{E}_{\rho \sim \mathcal{D}}[Re[Tr(\mathbf{U}_1^\dagger \mathbf{U}_2\rho)]]$. The inner products also induce corresponding $L_2$ norms: $\| \mathbf{M} \|_{\mathcal{D}} = \sqrt{\langle \mathbf{M}, \mathbf{M} \rangle_{\mathcal{D}}}$. As the magnitude of this norm can change with the dimension, we introduce the quantity $C_{\max}$ to denote the maximum value a query can take for any target observable in the *qCSQ* model, i.e., $C_{\max} = \max_{\mathbf{M}:\|\mathbf{M}\| \le 1} \| \mathbf{M} \|_{\mathcal{D}}^2$. For fair comparison, we quantify noise tolerances and hardness bounds with respect to $C_{\max}$. Note that for the *qUSQ* model $C_{\max} = 1$, but in the *qCSQ* model, $C_{\max}$ can decay with the number of qubits under for example the Haar distribution of inputs.

A statistical query algorithm learns a function class if it can output a unitary or observable that is close to any target in that class.

**Definition 5.** (*qCSQ*/*qUSQ* learning of hypothesis class). A given algorithm using only statistical queries to *qCSQ* (*qUSQ*) successfully learns a hypothesis class $\mathcal{H}$ consisting of observables **M**, $\|\mathbf{M}\| \le 1$ (unitaries **U**) up to $\epsilon$ error if it is able to output an observable **O** (unitary **V**) which is $\epsilon$-close to the unknown target observable $\mathbf{M} \in \mathcal{H}$ ($\mathbf{U} \in \mathcal{H}$) in the $L_2$ norm, i.e., $\| \mathbf{M} - \mathbf{O} \|_{\mathcal{D}} \le \epsilon$ ($\| \mathbf{U} - \mathbf{V} \|_{\mathcal{D}} \le \epsilon$).

The statistical query dimension quantifies the complexity of a hypothesis class $\mathcal{H}$ and is related to the number of queries needed to learn functions drawn from a class, as summarized in Theorem 7.

**Definition 6.** (Statistical query dimension[1,54]). For a distribution $\mathcal{D}$ and concept class $\mathcal{H}$ where $\| \mathbf{M} \|_{\mathcal{D}}^2 \le C_{\max}$ for all $\mathbf{M} \in \mathcal{H}$, the statistical query dimension ($SQ\text{-}DIM_{\mathcal{D}}(\mathcal{H})$) is the largest positive integer $d$ such that there exists $d$ observables $\mathbf{M}_1, \mathbf{M}_2, \ldots, \mathbf{M}_d \in \mathcal{H}$ such that for all $i \ne j$: $|\langle \mathbf{M}_i, \mathbf{M}_j \rangle_{\mathcal{D}}| \le C_{\max}/d$.

**Theorem 7.** (Query complexity of learning[1,2]). Given a distribution $\mathcal{D}$ on inputs and a hypothesis class $\mathcal{H}$ where $\| \mathbf{M} \|_{\mathcal{D}}^2 \le C_{\max}$ for all $\mathbf{M} \in \mathcal{H}$, let $d = SQ\text{-}DIM_{\mathcal{D}}(\mathcal{H})$ be the statistical query dimension of $\mathcal{H}$. Any *qCSQ* or *qUSQ* learner making queries with tolerance $C_{\max}\tau$ must make at least $(d\tau^2 - 1)/2$ queries to learn $\mathcal{H}$ up to error $C_{\max}\tau$.

Since our setting differs slightly from the standard classical setting[1,2], we include a proof of the above in Supplementary Note 3. For example, if the hypothesis class is rich enough to be able to express

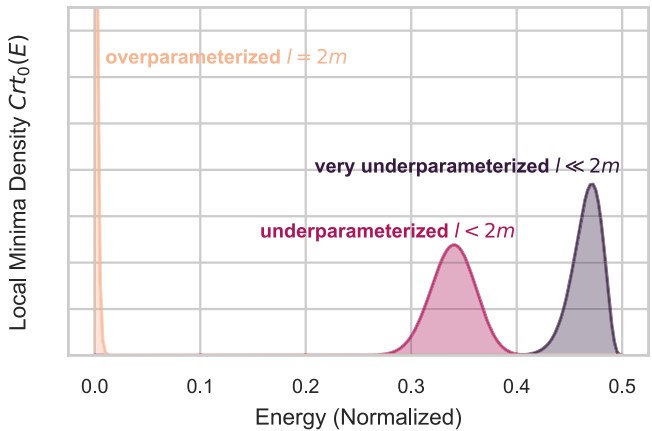

**Fig. 4 | Characteristic distribution of local minima.** Plot of the asymptotic distribution of local minima of WHRFs with $m$ degrees of freedom on the $l$-torus in: the extremely underparameterized regime, where $l \ll 2m$; the moderately underparameterized regime, where $l$ is a finite fraction of $2m$; and at the critical overparameterization regime, where $l = 2m$. Here, the energy is scaled and shifted as per (9) so that global minima have zero energy. In the underparameterized regime, only a fraction $\sim \exp(-m)$ of the critical points are within any constant additive error of the global minimum. In the overparameterized regime, local minima are exponentially concentrated at the global minimum.

any $n$-qubit Pauli observable, then the statistical query dimension of that class is at least $4^n$ over the Haar distribution of inputs since Pauli observables are all orthogonal. This forms the basis for our resulting proofs of hardness, summarized in Table 2 and proved in Supplementary Note 3.

Analogous to work in classical machine learning[28], one can perform noisy gradient descent as a series of statistical queries. As an example, consider the task of learning a target Hamiltonian **M** by constructing a variational Hamiltonian $\mathbf{H}(\boldsymbol{\theta}) = \mathbf{U}(\boldsymbol{\theta})^\dagger \mathbf{H} \mathbf{U}(\boldsymbol{\theta})$ with parameterized Pauli rotations and minimizing the mean squared error between expectations of **M** versus $\mathbf{H}(\boldsymbol{\theta})$ over a distribution of states $\mathcal{D}$. Our loss function is

$$\mathcal{L}(\boldsymbol{\theta}) = \mathbb{E}_{\rho \sim \mathcal{D}}\left[(Tr[\mathbf{M}\rho] - Tr[\mathbf{H}(\boldsymbol{\theta})\rho])^2\right]. \quad (16)$$

The parameter shift rule[55] provides a means to calculate the partial derivative of a function $f(\mu)$ with respect to a parameter $\mu$ applied as a parameterized quantum gate $e^{-i\mu\mathbf{G}}$ by calculating the function itself at two shifted coordinates. For example, for parameterized Pauli gates ($\mathbf{G} \in \frac{1}{2}\{\mathbf{Z},\mathbf{X},\mathbf{Y}\}$), this takes the form:

$$\frac{\partial}{\partial\mu}f(\mu) = \frac{1}{2}\left[f\left(\mu + \frac{\pi}{2}\right) - f\left(\mu - \frac{\pi}{2}\right)\right]. \quad (17)$$

By applying the parameter shift rule[55], we can evaluate the gradient of the loss with respect to parameter entry $\theta_i$ as

$$\frac{\partial}{\partial\theta_i}\mathcal{L}(\boldsymbol{\theta}) = \mathbb{E}_{\rho \sim \mathcal{D}}\left[(Tr[\mathbf{H}(\boldsymbol{\theta})\rho] - Tr[\mathbf{M}\rho])(Tr[\mathbf{H}(\theta^+)\rho] - Tr[\mathbf{H}(\theta^-)\rho])\right], \quad (18)$$

where $\boldsymbol{\theta}^+$ and $\boldsymbol{\theta}^-$ are the values of the parameters shifted at the $i$th entry according to the parameter shift rule for the gradient. The quantity $\mathbb{E}_{\rho \sim \mathcal{D}}\left[Tr[\mathbf{H}(\boldsymbol{\theta})\rho](Tr[\mathbf{H}(\theta^+)\rho] - Tr[\mathbf{H}(\theta^-)\rho])\right]$ can be directly evaluated without statistical queries, and the quantity $\mathbb{E}_{\rho \sim \mathcal{D}}\left[Tr[\mathbf{M}\rho](Tr[\mathbf{H}(\theta^+)\rho] - Tr[\mathbf{H}(\theta^-)\rho])\right]$ can be evaluated using 2 statistical queries to $qCSQ$ where the tolerance $\tau$ accounts for the noise in the estimate.

As a second example, this time in the unitary compiling setting of $qUSQ$, we can evaluate the commonly used procedure of measuring the inner product or average fidelity of $n$-qubit states between

a target unitary $\mathbf{U}_*$ and a variationally chosen unitary $\mathbf{V}(\boldsymbol{\theta})$ using statistical queries analogous to a swap test on actual quantum hardware[39,56–58]. With slight abuse of notation, let $|\phi\rangle \sim \mathcal{D}$ denote a distribution over pure states which forms a 2-design. Then via averaging over 2-designs (see Supplementary Note 3 for details), the average fidelity equals

$$\mathbb{E}_{|\phi\rangle \sim \mathcal{D}}\left[F(\mathbf{U}_*|\phi\rangle, \mathbf{V}(\boldsymbol{\theta})|\phi\rangle)\right] = \mathbb{E}_{|\phi\rangle \sim \mathcal{D}}\left[|\langle\phi|\mathbf{V}(\boldsymbol{\theta})^\dagger\mathbf{U}_*|\phi\rangle|^2\right] = \frac{2^{-n}|Tr(\mathbf{V}(\boldsymbol{\theta})^\dagger\mathbf{U}_*)|^2 + 1}{2^n + 1}. \quad (19)$$

Note that the key quantity $|Tr(\mathbf{V}(\boldsymbol{\theta})^\dagger\mathbf{U}_*)|^2 = \text{Re}[Tr(\mathbf{V}(\boldsymbol{\theta})^\dagger\mathbf{U}_*)]^2 + \text{Re}[iTr(\mathbf{V}(\boldsymbol{\theta})^\dagger\mathbf{U}_*)]^2$ can be evaluated up to a desired tolerance using statistical queries $qUSQ(\mathbf{V}(\boldsymbol{\theta}), \tau)$ and $qUSQ(i\mathbf{V}(\boldsymbol{\theta}), \tau)$.

One important caveat is that in the SQ setting, learning must succeed for all values of the query within the given tolerance $\tau$. Noise in quantum settings, which can arise from sampling a finite data set, gate error, state preparation error, measurement sampling noise, or other means does not exactly coincide with the assumed tolerance of an SQ model. Nevertheless, though noise during optimization may appear unnatural in classical settings, such noise in quantum settings is rather endemic and the SQ model allows one to rigorously analyze the complexity of learning in the presence of noise.

### The loss landscapes of Wishart hypertoroidal random fields
The loss landscapes of Wishart hypertoroidal random fields (WHRFs; see Supplementary Note 4 for a brief review) are known[11] to exhibit a computational phase transition governed by the order parameter

$$\gamma = \frac{l}{2m}, \quad (20)$$

called the overparameterization ratio. Here, $l$ is the number of parameters of the WHRF, and $m$ its degrees of freedom (see Supplementary Note 4). When $\gamma \ll 1$ (the underparameterized regime), WHRFs exhibit poor local minima and thus are essentially untrainable; when $\gamma \geq 1$ (the overparameterized regime), however, essentially all local minima of a WHRF are close to the global minimum in function value. More specifically, when $\gamma = o\left(\frac{1}{\log(n)}\right)$, a superpolynomially small (in $n$) fraction of the local minima are within any constant additive energy error to the global minimum. When restoring units to the variational risk of (9), this is an error extensive in the problem size. The asymptotic expression of the distribution of local minima is also known, which is given by (up to a normalization factor):

$$Crt_0(E) \sim e^{-mE}E^{m-l/2}(1 - 2E)^l \quad (21)$$

for the density of local minima at any given energy $0 \leq E \leq \frac{1}{2}$, in units of the mean eigenvalue of **H** (shifted such that the global minimum is at $E = 0$). Representative plots of this distribution in various parameterization regimes are shown in Fig. 4.

This distribution of local minima is calculated from the joint distribution of the WHRF function value, its gradient, and its Hessian. Thus, by demonstrating the convergence of this joint distribution in the variational loss functions we consider to the analogous distribution in WHRFs at a sufficient rate, we are able to show the same phase transition occurs in variational loss functions. Our full proof is given in Supplementary Note 4.

### Data availability
The processed data generated and analyzed for this study are available at https://github.com/bkiani/Beyond-Barren-Plateaus and ref. 59.

### Code availability
The code used for the current study is available at ref. 59.

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

## Acknowledgements

The authors thank Giacomo De Palma, Tongyang Li, Seth Lloyd, Milad Marvian, Quynh T. Nguyen, and Agnes Villanyi for helpful feedback and discussions. E.R.A. was supported by the National Science Foundation Graduate Research Fellowship Program under Grant No. 4000063445. B.T.K. was supported by the MIT Energy Initiative fellowship.

## Author contributions

E.R.A. and B.T.K. both wrote this manuscript and formulated the original project ideas, contributed to the proofs, and performed numerical experiments.

## Competing interests

The authors declare no competing interests.
