## [Peer Review File · Nature Communications]

Quantum Variational Algorithms Are Swamped With TrapsREVIEWER COMMENTS

Reviewer #1 (Remarks to the Author):

SEE BELOW

Reviews Comments of

Beyond Barren Plateaus: Quantum Variational Algorithms Are Swamped With Traps

Eric R. Anschuetz
Bobak T. Kiani

Journal: *Nature Communication*,

1. The main work of this manuscript

This paper by Eric R. Anschuetz, Bobak T. Kiani mainly discuss a rigorous analysis about the trainability of a class of variational quantum algorithms. The manuscript is well-written and mainly leverages the established techniques from statistical learning theory and random matrix theory to analysis the training issue of shallow quantum circuit even without barren plateaus. Under quantum statistical query framework, they pointed out it typically requires exponentially many queries for VQAs model to learn. They also discuss that there are exponential poor local minima with system size increasing from loss landscape perspective. In summary, authors point out the training of VQAs with shallow quantum circuit is typically infeasible.

2. Comments #1

In page 3. Quantum machine learning algorithms have been a focus of intense research effort as potential use-cases for noisy, intermediate-scale quantum ... as we summarize in Section 2.2..

This section introduces the basic concepts of classical machine learning like population risk, generalization error that are not so related to the context of the manuscript. Thus, it is suggested to move them to the appendix.

3. Comments #2

In page 9, it shows that the loss landscape of are exhibit a computational phase transition that is governed by the overparametrization ratio.

Recently, a few study discuss about the overparametrization of the VQAs, such as arXiv:2109.11676 and arXiv:2205.12481, and give the overparametrization threshold, like maximum achievable rank of quantum fisher information or function of effective dimension and effective spectral ratio. In the manuscript, authors directly give the overparameterization region $\gamma \leq 1$. it is not obviously how to determine the value of overparameterization threshold.

4. Comments #3

In page 8 table 2, it shows some hardness of SQ learning of different settings

what about query complexity of the setting, $L \ll n$, both single and two qubit gates, that is more general case used in VQAs such as HEA.

Reviewer #2 (Remarks to the Author):

In the paper titled “Beyond Barren Plateaus: Quantum Variational Algorithms Are Swamped With Traps,” the authors study trainability of the variational quantum algorithms (VQAs) via the statistical query framework. They show that the noisy optimization of a wide variety of quantum models is impossible with a sub-exponential number of queries. The existing untrainability results in the literature either concern deep VQAs or nonlocal models. While the barren plateau in the former case is due to vanishing gradients, the latter involves poor local minima. No rigorous results were known for shallow, local quantum models with local cost functions. The authors show that the shallow, local quantum models with local cost functions are not trainable, especially when a good initialization is missing or the model does not exhibit a high amount of symmetry. Further, they extend the techniques from reference [3] to show negative trainability results for local shallow circuits that do not suffer from barren plateau. Furthermore, they also discuss trainability phase transition as number of parameters are increased. The result looks interesting and promising. I have following comments and questions.

- (1) The meaning of dimension in Table 1 is not clear.
- (2) The section 3.1 is terse and needs to be expanded. More explanation is needed for the calculation of the gradient.
- (3) In terms of techniques, the authors use either the existing tools in the literature or ideas from reference 3. What are the novel contributions of this work (in terms of techniques)? Maybe the authors should add a section explaining new techniques proposed in their current work.
- (4) In Table 2, exponentially small tolerance maybe too much small.
- (5) Based on the discussion on page 6, it seems the hardness of learnability in the statistical query setting does not imply hardness of learning for any algorithm. It seems the results in the paper are just evidence and not the final word. This makes me wonder if the title of the paper makes sense. Am I missing something?

I will make a judgement once I see the revised version of the manuscript and their response to my questions/comments.

Reviewer #3 (Remarks to the Author):

The paper tackles optimization problems in variational quantum algorithms. This is an important area of research. It is expected that near-term noisy quantum computers will continue to be used in a hybrid way. That is, a quantum computer will be utilized to estimate some predefined cost function and a classical computer will analyze that data and return a different circuit with lower value of the cost function. The minimum of a cost function is then found by running this procedure in a loop. The original problem is thus converted into finding the minimum of a cost function. Difficult problems typically result in hard-to-optimize landscapes. Past literature analyzed, to a large extent, only one such problem, called a barren plateau issue. It has been shown that some cost functions coupled with random initialization lead to a flat landscape that requires exponential resources to navigate.

The authors concentrate on a different issue that is affecting variational quantum algorithms. They address the following important question: assuming a given optimization setup is free from barren plateau issues, are there other problems that may prevent successful training? The answer to that question, provided by the authors, is “yes” and those problems are related to the abundance of low-

quality local minima. This sets the paper apart from mainstream research in the field that is still concerned with barren plateaus and corresponding mitigation methods.

While I am very much in favor of accepting this paper to Nature Communications, I would like the authors to address two points: (i) layout of the article and choice of material, and (ii) choice of numerical experiments. I detail my concerns below.

(i) The layout of the manuscript does not match Nature Communications style, which is: 1. Introduction, 2. Results, 3. Discussion, 4. Methods. Right now, the paper goes back and forward between background information on a given topic, new methods that the authors are developing, and results. It makes the paper rather hard to follow, goals and motivations change throughout the text and it is not clear how the logic of the article evolves. The sentence “We now move on to discussing the trainability of variational quantum algorithms in another light.” suggests that the paper could be divided into two, each with a much clearer message.

The above-mentioned style will force the authors to rethink and probably reduce the material. Many readers will find the material covered in “Learning in the Statistical Query Framework” section to be disconnected from the rest of the paper. Is that part actually needed? Could it be compressed down to a clear message, something along the lines of the last paragraph of Section 3, while the details delegated to Supplemental Information (SI) or used to write another paper?

The authors should also rethink the balance between the material covered in the main text and in the SI. The paper should defend itself based on the main text alone. SI should not contain any discussion relevant to the main points of the article. For example, I would move Section 7 of SI to the main text (some of it is described in Conclusions but should be upgraded to a subsection of Discussion in the new format). “Reason for optimism” announced by the authors in the abstract should not be buried somewhere on page 21 of SI.

(ii) The authors discuss three numerical examples to illustrate their theoretical findings: (a) select a random QCNN and recover it with another QCNN, (b) learn a quantum state generated by random quantum circuit, and (c) find the ground state of non-interacting Hamiltonian randomized by depth-L quantum circuit. Those examples are among the hardest tasks that one can think of using hybrid quantum-classical algorithms for. It is not surprising that the authors see optimization problems there. For example, in (c), in the limit of large L, one is tasked with finding the ground state of a random Hamiltonian - a problem that is prohibitively difficult. Surely, those limits are not achieved in their studies. Also, those examples illustrate potential problems very well. However, they are also artificial. It is hard to imagine that future quantum advantage experiments will be contingent on learning random QCNN. I believe that the paper will be much more impactful if the authors consider more practical problems. For example, instead of working with random Hamiltonian, analyze trainability issues of a local one, inspired by problems in material science, something that researchers may actually consider running on a quantum computer. I would also perform a more realistic task with QCNN (say, classification) as opposed to performing random “teacher-student learning”. The paper will be much stronger if the authors manage to find optimization problems in tasks that the field considers good candidates for quantum advantage.

Reviewer #4 (Remarks to the Author):

SEE BELOW

1 Overview

This paper studies properties and limitations of variational quantum algorithms where the ansatz is a shallow depth quantum circuit. The following are their main contributions.

2 Contributions

2.1 Connections to the statistical query model

They relate trainability of their shallow depth ansatz with the quantum analogue of a “statistical query learning model” from learning theory. The oracle, for query purposes, is a noisy Hamiltonian. The goal is to keep querying a series of noisy oracles to finally distill the required information from the target Hamiltonian.

They relate the statistical query learning model with gradient descent based optimization techniques (section 3.1.) What they show is, one run of gradient descent for certain Hamiltonians, where the gradient is computed by the parameter shift rule, corresponds to a constant number of queries to some noisy Hamiltonian.

So, for those Hamiltonians, by bounding the number of queries needed to get useful information out of the target Hamiltonian, we can bound the convergence rate of any gradient descent based optimization technique. They show that for the shallow depth case, the number of queries is lower bounded by $\Omega(2^n)$, where n is the number of qubits, which rules out efficient optimizers based on gradient descent.

Note that this mapping works only for certain classes of Hamiltonians. In the last paragraph of section 3.2, they provide an instance where this mapping to the statistical query model breaks down.

2.2 Connections to WHRFs

They relate the cost function landscapes of some shallow depth ansatzes with “Wishart hypertoroidal random fields (WHRF).” More specifically, they look at “locally scrambled ansatzes:” ansatzes where there are pockets such that the landscape looks “flat” in those pockets.

The landscape of WHRFs were studied in the literature before. So, results from there apply to this setting too. Specifically, WHRFs have very few local minimas within a small neighborhood of the global minima. What this means is, either the optimization algorithm luckily finds the global minima by virtue of a good initial guess, or it gets stuck in a local minima that is far from the actual solution. They support their claims with examples (section 4.2) and numerical simulations (section 4.3).

3 Comments

3.1 Connections to previous work

The authors reference a recent paper by John Napp, in reference 45. Other recent works, for example [DNS⁺21], have also studied the output distribution of shallow depth random quantum circuits extensively.

3.1.1 Concentration of the output distribution

What we know is that the output distribution “concentrates:” there is some global maxima/minima, and a very suppressed cost landscape around that point. Moreover, if the observable we care about is written as

$$O = \sum_i \alpha_i P_i, \tag{1}$$

where each P_i is a Pauli operator, then the suppression, for each P_i , is inverse exponential in the weight (number of non-identity terms in the tensor product representation) of P_i and the depth of the prepared ansatz (see Corollary 1, 2, and 3 of John Napp’s paper.) For shallow depth circuits, for each P_i , the dominant factor in the exponential is the weight of the term.

3.1.2 Connections to present work

I did not fully grasp the similarities and dissimilarities between the present paper and previous works like these. What these works say is, the cost landscape is essentially “flat” for any term with more than $O(\log n)$ weight. So, there is no hope to find any local minima there.

- Are the authors claiming that even for terms with less than $O(\log n)$ weight, the cost landscape is essentially still “flat” – few local minimas can be found there as well?
- So, is the conclusion that there are very few local minimas in general, overall, regardless of weight?

If so, a proper justification of such claims would be very helpful.

3.2 Clear distinction between worst and average case

The authors seem to talk about “worst case” analysis and worst case statistical query bounds for the first case and an “average case” analysis for the second case.

If so, this should be made clear somewhere, with, perhaps, a discussion on what is known about average case query bounds for the statistical learning model and how their two approaches to argue untrainability are related. Otherwise, it is hard to parse the “Our results” row of Table 1. For average case untrainability, does the second approach, qualitatively, already prove what the first approach is gunning for (difficulty of training these models)?

3.3 Clearer depiction of edge cases

A more detailed analysis on how general their statistical query lower bounds are would be helpful. In other words, what are the Hamiltonians for which they break down – are these edge cases physically motivated or artificially concocted? For Hamiltonians where the query lower bounds break down, and for a randomized ansatz, does the mapping to WHRFs hold at least?

4 Conclusion

The mapping to statistical query lower bounds and the mapping to WHRFs are both very interesting and technically novel. However, I am not yet convinced that they are conceptually novel or whether they are majorly rediscovering what was known before about shallow depth random circuits. I am also not convinced that the first approach (mapping to a statistical query model) is of independent interest qualitatively and does not get subsumed by the second approach (mapping to WHRFs.)

In light of these points, my recommendation is that the current version of the manuscript is not suitable for *Nature Communications*.

References

- [DNS⁺21] Abhinav Deshpande, Pradeep Niroula, Oles Shtanko, Alexey V. Gorshkov, Bill Fefferman, and Michael J. Gullans. Tight bounds on the convergence of noisy random circuits to the uniform distribution, 2021.

Response to the Reviewers

We thank the reviewers for their comments on our manuscript, and the generally positive feedback. We agree with the reviewers that the previous organization of the manuscript made the relative merits of both our statistical query (SQ) learning results and our loss landscape results less clear.

In light of these reviews, we would like to address the reviewers’ questions and concerns below. Before proceeding, we make note of the following primary changes to the manuscript which address comments from both the editor and the reviewers:

- We have restructured the manuscript to the style of Nature Communications. This has allowed us to move relevant details from the Supplemental Information to the main text, while also allowing us to clarify our results by moving mathematical details to the Methods. We believe that this has greatly improved the structure and clarity of the manuscript.
- We have clarified how our techniques and results differ from previous results on the trainability of quantum machine learning algorithms. In particular, we have clarified in Sec. 2.3 that our techniques for showing that shallow variational quantum loss functions approach those of Wishart hypertoroidal random fields (WHRFs)—and thus inherit their poor training landscapes—are novel. Previous research in this direction has only been able to prove that unphysical, extremely nonlocal ansatzes converge to WHRFs, as previous techniques have relied on the global scrambling of the ansatz to prove this convergence. We are able to show that even physically realistic local ansatzes which do *not* have this global scrambling property converge to WHRFs. We do this by using results from [6] to show that, though such ansatzes do not scramble globally in shallow depth, they *approximately* scramble *locally*. We then construct a bound on the error in the joint characteristic function of the loss function and its first two gradients to bound the error in distribution that is incurred by this scrambling being only approximate, and then show using properties of local Haar random gates and the locality of the problem Hamiltonian that this suffices to prove convergence of these random variables to those of a WHRF. We then show that this convergence is enough to demonstrate that the distribution of local minima converges to that of a WHRF. We have now clarified these distinctions in detail in Sec. 2.3. We also now clarify in Sec. 2.2 that the novelty of our statistical query lower bounds come from bounding the statistical query dimension of the function classes we consider.
- We have clarified the relative strengths of the SQ learning framework when compared with the loss landscape analysis. In particular, we now emphasize in Sec. 2.2 that the SQ results are very general no-go results that hold for any problem class; however, they require the assumption of noise in the training procedure. Our loss landscape results in Sec. 2.3 are complementary in the sense that they do *not* require this assumption of noise in the training procedure to demonstrate untrainability, but they only hold in “typical” cases; this is now clarified in Sec. 2.3.

Reviewer 1

We thank the reviewer for complementing the exposition and rigor of the manuscript. We have addressed their comments below.

Reviewer Point P 1.1 — [Sec. 2.1] introduces the basic concepts of classical machine learning like population risk, generalization error that are not so related to the context of the manuscript. Thus, it is suggested to move them to the appendix.

Reply: Following the reviewer’s recommendation, we have considerably shortened what was Sec. 2.1 and have moved most of the corresponding details to the Supplementary Information. We have merged the remaining discussion with what was Sec. 2.2 as a brief overview of the setting of quantum machine learning algorithms we study.

Reviewer Point P 1.2 — Recently, a few study discuss about the overparametrization of the VQAs, such as arXiv:2109.11676 and arXiv:2205.12481, and give the overparametrization threshold, like maximum achievable rank of quantum fisher information or function of effective dimension and effective spectral ratio. In the manuscript, authors directly give the overparameterization region $\gamma \leq 1$. it is not obviously how to determine the value of overparameterization threshold.

Reply: We believe both of these papers are relevant and have cited them in our manuscript. Indeed, our main result when studying the loss landscapes of local variational quantum algorithms (VQAs) is the calculation of γ , which then allows us to determine where the phase transition in trainability occurs. We do this via Theorem 2.1—which shows that the loss functions of the class of VQAs we study converge to those of WHRFs—with the explicit value for γ given in Eq. (20). The full proof of the Theorem is given in the Supplemental Information. Unlike the references cited, we are able to explicitly find the value of γ and demonstrate the associated phase transition in trainability by considering an explicit model for local VQAs, as given in Eq. (9) and the surrounding text.

Reviewer Point P 1.3 — what about query compelxity of the setting, L, n , both single and two qubit gates, that is more general case used in VQAs such as HEA

Reply: We already have calculated the query complexity of this setting, with the result summarized in the third line of Table 2 (Corollary 3.11). Though we previously only stated “two qubit gates,” note that single qubit gates can be considered two qubit gates via tensor product with the identity. We have now clarified this in Table 2.

Reviewer 2

We thank the reviewer for their comments and questions, which we address below.

Reviewer Point P 2.1 — The meaning of dimension in Table 1 is not clear.

Reply: We agree that the usage of “Dimension” was unclear—in the caption of Table 1, we have now specified it refers to the locality structure of the ansatzes being studied. We now clarify the “Worst case” column of Table 1 in its caption as well.

Reviewer Point P 2.2 — The section 3.1 is terse and needs to be expanded. More explanation is needed for the calculation of the gradient.

Reply: After reorganizing the paper to fit the Nature Communications style, we have moved much of what was in Sec. 3.1 to the Methods section, including more examples and details. We now are more explicit with details of the associated calculations, including equations used to obtain the gradient from the parameter shift rule.

Reviewer Point P 2.3 — In terms of techniques, the authors use either the existing tools in the literature or ideas from reference 3. What are the novel contributions of this work (in terms of techniques)? Maybe the authors should add a section explaining new techniques proposed in their current work.

Reply: Reference 3 (cited as [1] in this response) demonstrated that certain variational quantum algorithms (VQAs) asymptotically approach a certain class of random fields (Wishart hypertoroidal random fields (WHRFs)) that have poor distributions of local minima, implying that such VQAs also have poor local minima and thus are difficult to train. However, the ansatzes considered in [1] are very physically unrealistic—in particular, they considered very nonlocal ansatzes with high weight Pauli rotations. This was needed to show that the ansatz scrambled sufficiently even at shallow depths to demonstrate convergence to WHRFs. Our main contribution in Sec. 2.3 is demonstrating that more realistic (i.e. d -dimensional, local) ansatzes also asymptotically approach this same class of random field. However, doing this requires completely different techniques from [1], since these

models do not scramble at shallow depths. Instead, we show that such ansatzes *approximately* scramble *locally* at shallow depths. We then use novel techniques to show that this approximate, local scrambling is sufficient to show convergence of the loss function and its first two derivatives to those of WHRFs, which we then show suffices to demonstrate convergence in the distributions of local minima. We have added more discussion to the beginning of Sec. 2.3 to clarify our novel contributions here. We also now clarify in Sec. 2.2 that the novelty of our statistical query lower bounds come from bounding the statistical query dimension of the function classes we consider.

Reviewer Point P 2.4 — In Table 2, exponentially small tolerance maybe too much small.

Reply: Indeed, the reviewer is correct that the minimum tolerance required for our proofs to hold is very small (exponentially small in the problem size); however, our results hold for all tolerances that are *at least* this value. This is one major reason why learning is so hard in this setting. Though one can avoid our results in Table 2 with sufficiently small tolerance, this tolerance would have to be exponentially small in the problem size. For typical optimizers which sample gradients on a quantum computer, this alone implies that exponentially many samples are required to measure the gradient to the required precision to avoid our no-go results. We have clarified this in the main text, in Sec. 2.2.

Reviewer Point P 2.5 — Based on the discussion on page 6, it seems the hardness of learnability in the statistical query setting does not imply hardness of learning for any algorithm. It seems the results in the paper are just evidence and not the final word. This makes me wonder if the title of the paper makes sense. Am I missing something?

Reply: Statistical query (SQ) lower bounds directly apply to algorithms whose basic steps reduce to statistical queries. This includes most algorithms practically used in the training of variational quantum algorithms (VQAs), including gradient descent (and other first order optimization methods), BFGS (and other second order optimization methods), and all optimization methods that train via queries to the loss function. In our manuscript, we state that there exist training algorithms beyond these settings, but they are often constructed with some specific knowledge of the physical problem at hand in order to be efficiently trainable. As an example, we include in the main text references to [3, 7, 5], which construct methods for learning Pauli channels efficiently outside of the SQ framework, but which rely on specific properties of Pauli channels in order to achieve this. We have added more discussion both in Sec. 2.2 and the Methods (Sec. 4.1.1) clarifying and emphasizing the broad applicability of the SQ framework. These no-go results in the SQ framework, along with our proof on the proliferation of poor local minima (“traps”) in VQAs, we believe justify our title.

Reviewer 3

We thank the reviewer for their favorable comments about our manuscript and recommending acceptance. We have addressed their specific comments and questions below.

Reviewer Point P 3.1 — The layout of the manuscript does not match Nature Communications style, which is: 1. Introduction, 2. Results, 3. Discussion, 4. Methods. Right now, the paper goes back and forward between background information on a given topic, new methods that the authors are developing, and results. It makes the paper rather hard to follow, goals and motivations change throughout the text and it is not clear how the logic of the article evolves. The sentence “We now move on to discussing the trainability of variational quantum algorithms in another light.” suggests that the paper could be divided into two, each with a much clearer message.

The above-mentioned style will force the authors to rethink and probably reduce the material. Many readers will find the material covered in “Learning in the Statistical Query Framework” section to be disconnected from the rest of the paper. Is that part actually needed? Could it be compressed down to a

clear message, something along the lines of the last paragraph of Section 3, while the details delegated to Supplemental Information (SI) or used to write another paper?

The authors should also rethink the balance between the material covered in the main text and in the SI. The paper should defend itself based on the main text alone. SI should not contain any discussion relevant to the main points of the article. For example, I would move Section 7 of SI to the main text (some of it is described in Conclusions but should be upgraded to a subsection of Discussion in the new format). “Reason for optimism” announced by the authors in the abstract should not be buried somewhere on page 21 of SI.

Reply: We agree with the reviewer that the main message of our results were lost in mathematical details previously included in the main text. We have thus followed the reviewer’s suggestion, and have restructured the paper into the Nature Communications format. We have moved many of the details of both the statistical query framework and Wishart hypertoroidal random fields (WHRFs) to the Methods section and the Supplemental Information. In doing so, we have also made more clear the connections between our results in the statistical query learning framework and our results on the typical loss landscapes of shallow variational quantum algorithms (VQAs). The former gives very general no-go results when training in the presence of noise. The latter then shows that even without noise, for *typical* VQA instances, these models remain untrainable by directly studying the typical loss landscapes of such VQA instances. We believe the changes we have made to the manuscript make this main message of the Results section clearer. Following the reviewer’s suggestions, we have also moved important implications and interpretations of our results that were previously in the Supplemental Information—such as Sec. 7 of the Supplemental Information—to the main text, leaving only details of the proofs and numerics (and some supplemental numerics) for the Supplemental Information.

Reviewer Point P 3.2 — The authors discuss three numerical examples to illustrate their theoretical findings: (a) select a random QCNN and recover it with another QCNN, (b) learn a quantum state generated by random quantum circuit, and (c) find the ground state of non-interacting Hamiltonian randomized by depth- L quantum circuit. Those examples are among the hardest tasks that one can think of using hybrid quantum-classical algorithms for. It is not surprising that the authors see optimization problems there. For example, in (c), in the limit of large L , one is tasked with finding the ground state of a random Hamiltonian - a problem that is prohibitively difficult. Surely, those limits are not achieved in their studies. Also, those examples illustrate potential problems very well. However, they are also artificial. It is hard to imagine that future quantum advantage experiments will be contingent on learning random QCNN. I believe that the paper will be much more impactful if the authors consider more practical problems. For example, instead of working with random Hamiltonian, analyze trainability issues of a local one, inspired by problems in material science, something that researchers may actually consider running on a quantum computer. I would also perform a more realistic task with QCNN (say, classification) as opposed to performing random “teacher-student learning”. The paper will be much stronger if the authors manage to find optimization problems in tasks that the field considers good candidates for quantum advantage.

Reply: We thank the reviewer for pointing out some areas of confusion and weaknesses of the simulations. Before addressing the request for more realistic experiments, we want to clarify that in the random Hamiltonian experiments labeled as (c) by the reviewer, the depth L used in constructing the problem Hamiltonian was fixed at $L = 4$. We only scaled either the number of qubits in the Hamiltonian or the number of layers in the ansatz used to learn the ground state of this Hamiltonian. That way, the difficulty of the problem was fixed and we studied the impact of having richer ansatzes or more qubits. We have since clarified this in the main text, by using L^* when referring to the depth of the conjugation circuit of the Hamiltonian, and L when referring to the ansatz depth (such that overparameterization occurs when $L > L^*$). We also now clarify in Sec. 2.4.1 that the QCNN teacher-student learning task is a classification task, just one constructed by a teacher QCNN such that perfect classification accuracy in the student is known to be achievable.

Our goal in the experiments was to look at settings where we know a global minimum exists that solves the problem. That way, it is straightforward to disentangle challenges with optimization, generalization, and approximation errors. Nonetheless, we agree with the reviewer that performing simulations on more “realistic” experiments would be useful to the reader and further reinforce our main findings. To that end, we have included in

the Supplemental Materials simulations of using a variational quantum eigensolver (VQE) to find the ground state of a 2D Heisenberg XYZ model. There, many of the same findings are apparent as in our previous simulations. We find that finding the ground state is generally difficult at shallow depths due to the poor VQE loss landscape (consistent with our findings), and at larger depths optimization remains difficult as barren plateaus develop in the loss landscape (indicated by the poor performance when the resolution of the gradient measurements is limited by shot noise).

Reviewer 4

We thank the reviewer for their nice comments about the technical novelty of our work. We also thank the reviewer for pointing out the relevant work [4], and have cited this in the current version of the manuscript.

Reviewer Point P 4.1 — I did not fully grasp the similarities and dissimilarities between the present paper and previous works like these. What these works say is, the cost landscape is essentially “flat” for any term with more than $O(\log n)$ weight. So, there is no hope to find any local minima there.

- Are the authors claiming that even for terms with less than $O(\log n)$ weight, the cost landscape is essentially still “flat” – few local minimas can be found there as well?
- So, is the conclusion that there are very few local minimas in general, overall, regardless of weight? If so, a proper justification of such claims would be very helpful.

...

The mapping to statistical query lower bounds and the mapping to WHRFs are both very interesting and technically novel. However, I am not yet convinced that they are conceptually novel or whether they are majorly rediscovering what was known before about shallow depth random circuits.

Reply: Previous papers (such as [8, 2, 9]) that demonstrate the existence of barren plateaus show that the variance of the gradient for deep variational models is exponentially small, making training on a quantum computer difficult in practice due to the large number of measurements needed to overcome the shot noise. We believe the reviewer is making reference to [2, 9]. The reviewer is correct; here, we study local (i.e. $O(1)$ weight terms) Hamiltonians with local ansatzes. We demonstrate analytically (and confirm numerically) that these ansatzes are untrainable, even when they are shallow. However, we show that they are untrainable because they have many local minima that are far from the global minimum, making the model essentially untrainable as optimizers will get stuck in these poor local minima. In fact, [9] previously showed that the models we consider do *not* suffer from barren plateaus, which demonstrates that this is a completely different untrainability phenomenon than the barren plateau results studied previously. We have stated these implications more clearly (and formally) with Corollary 2.2 in the main text, Corollary 4.4 in the Supplemental Information, and exposition surrounding both. We also now clarify in Sec. 2.4.2 that the numerical experiments shown in Fig. 3(b) agree with our theoretical results on the existence of many poor local minima for local, shallow ansatzes until the overparameterization threshold is reached.

Reviewer Point P 4.2 — The authors seem to talk about “worst case” analysis and worst case statistical query bounds for the first case and an “average case” analysis for the second case.

If so, this should be made clear somewhere, with, perhaps, a discussion on what is known about average case query bounds for the statistical learning model and how their two approaches to argue untrainability are related. Otherwise, it is hard to parse the “Our results” row of Table 1. For average case untrainability, does the second approach, qualitatively, already prove what the first approach is gunning for (difficulty of training these models)?

Reply: The statistical query (SQ) approach yields a set of very general no-go results, which state that *no algorithm* (that utilizes statistical queries) in the presence of even a very small amount of noise can optimize *any* given problem in any of the settings we consider (i.e. any of the settings listed in Table 2, which are very broad).

In other words, in the SQ setting we are able to prove results stronger than “average case” results, as we show it holds true for all problems. In the context of SQ learning, what is potentially “worst case” is the noise of the statistical queries. We have clarified this in the main text, and have made this less ambiguous. We have also changed the statement of “Average case” results in Table 1 to “Worst case” (and inverted the check marks in that column), to emphasize that unlike some previous results on the untrainability of quantum models, we do not rely on “worst case” inputs to the model. We have also now emphasized in Secs. 2.2 and 2.3 that the strength of the SQ framework is proving very strong no-go learning results when even a very small amount of noise is present in the optimization procedure; our results on the landscapes of variational algorithms, though they hold true when there is no noise in the optimization procedure, only hold in the average case.

Reviewer Point P 4.3 — A more detailed analysis on how general their statistical query lower bounds are would be helpful. In other words, what are the Hamiltonians for which they break down – are these edge cases physically motivated or artificially concocted? For Hamiltonians where the query lower bounds break down, and for a randomized ansatz, does the mapping to WHRFs hold at least?

Reply: The statistical query lower bounds are for any Hamiltonian within a very wide class of models (see Table 2 for examples of many settings in which our results hold). Where the query lower bounds break down are when we resort to algorithms that do not reduce to statistical queries, or are given noiseless access to the gradient and/or loss function. The landscape/WHRF results demonstrate that even in this scenario, the landscapes are (for typical instances) very unamenable to optimization due to the presence of many extremely poor local minima. We have added supporting text in Secs. 2.2 and 2.3 clarifying these points.

Reviewer Point P 4.4 — I am also not convinced that the first approach (mapping to a statistical query model) is of independent interest qualitatively and does not get subsumed by the second approach (mapping to WHRFs.)

Reply: The SQ framework allows us to prove very general no-go results for efficient learning in the presence of noise for any problem (assuming the problem is in one of the settings discussed in Table 2). Our loss landscape results—though they hold even when no noise is present in the optimization procedure—only hold for typical instances. We agree that our original version made this distinction less clear, so have added additional exposition to Secs. 2.2 and 2.3 to clarify these points.

We once again thank the reviewers for their comments and questions. After following their recommendations, we believe the manuscript is now much clearer and easier to follow, and also more clearly delineates the different settings in which our statistical query results hold (i.e. with noise, all problem instances) and in which our landscape results hold (i.e. without noise, typical problem instances). We have also clarified the major distinctions in technique and assumptions of these latter results when compared with [1], both here and in the main text.

References

- [1] Eric Ricardo Anschuetz. Critical points in quantum generative models. In *International Conference on Learning Representations*, 2022.
- [2] M. Cerezo, Akira Sone, Tyler Volkoff, Lukasz Cincio, and Patrick J. Coles. Cost function dependent barren plateaus in shallow parametrized quantum circuits. *Nat. Commun.*, 12(1):1791, 2021.
- [3] Senrui Chen, Sisi Zhou, Alireza Seif, and Liang Jiang. Quantum advantages for Pauli channel estimation. *Physical Review A*, 105(3):032435, 2022.
- [4] Abhinav Deshpande, Pradeep Niroula, Oles Shtanko, Alexey V. Gorshkov, Bill Fefferman, and Michael J. Gullans. Tight bounds on the convergence of noisy random circuits to the uniform distribution, 2021.

- [5] Aravind Gollakota and Daniel Liang. On the hardness of pac-learning stabilizer states with noise. *Quantum*, 6:640, 2022.
- [6] Aram Harrow and Saeed Mehraban. Approximate unitary t -designs by short random quantum circuits using nearest-neighbor and long-range gates, 2018.
- [7] Hsin-Yuan Huang, Richard Kueng, and John Preskill. Information-theoretic bounds on quantum advantage in machine learning. *Physical Review Letters*, 126(19):190505, 2021.
- [8] Jarrod R. McClean, Sergio Boixo, Vadim N. Smelyanskiy, Ryan Babbush, and Hartmut Neven. Barren plateaus in quantum neural network training landscapes. *Nat. Commun.*, 9(1):4812, 2018.
- [9] John Napp. Quantifying the barren plateau phenomenon for a model of unstructured variational ansätze, 2022.

REVIEWERS' COMMENTS

Reviewer #1 (Remarks to the Author):

The authors have satisfactorily addressed the comments and the paper is now suitable for publication.

Reviewer #2 (Remarks to the Author):

I went through the updated manuscript and the response file. The authors have answered four out of five concerns raised by me. However, I am still not convinced regarding point 2.3. In terms of techniques, the work seems a derivative of reference 1 and other existing tools in the literature. I am not sure if Nature Com requires the novelty of the technique as a pre-requisite for the publication. I will be neutral and leave it to the editor to decide. I am okay with acceptance, since most of my concerns have been addressed.

Reviewer #3 (Remarks to the Author):

The authors addressed all my concerns in a satisfactory manner. I recommend publication.

Reviewer #4 (Remarks to the Author):

SEE BELOW

Response to authors' comments

- The authors clarify the connection between previous results and their work in Corollary 2.2 of the main text (and also in the supplemental material.) The most relevant work is that of [Nap22]. There, it is shown how shallow depth random circuit ansatzes do not exhibit barren plateaus. In the present work, the authors prove that for shallow depth random circuit ansatzes, even though there aren't barren plateaus, the local minimas are far away from the global minima. So, such ansatzes are untrainable because there is a high probability of the optimizer getting stuck in a local minima.

However, although the authors' technical result is novel, the message is not. In [DNS⁺21, Nap22], it is proven that shallow depth random ansatzes exhibit a severe lack of "anti-concentration." In other words, the cost function is sharply "peaked" at certain points. Furthermore, [Nap22] proves that the peaks are inverse exponentially suppressed depending on the Hamming weight of the Pauli string under consideration. So, the strings with low Hamming weight (at most $O(\log n)$ Hamming weight) are the only type of strings that can exhibit sharp speaks. This *already* makes these types of cost functions mostly useless, because the ansatz is restricted to exploring only an exponentially small fraction of the full space.

So, even though barren plateaus are not exhibited, it was already known that such cost functions are essentially useless for optimization tasks.

- The authors clarify the strengths of their statistical query model in Sections 2.2 and 2.3 of the paper. Specifically, they clarify that for some regimes, the statistical query model gives worst case hardness results, which is stronger than arguing about average case hardness.

Once again, the technical novelty of the statistical query model and the hardness results the authors prove using that model is interesting. But, I am not sure how novel the messaging is conceptually. It is not clear to me why average case hardness analysis, using loss landscapes, is not already proof enough of the fact that shallow depth ansatzes are bad for optimization tasks. Although a worst case analysis is technically interesting, it does not add too much to the messaging, because such worst-case ansatzes may be inefficient to construct in the first place.

In light of these two issues, I do not recommend acceptance of the paper.

References

- [DNS⁺21] Abhinav Deshpande, Pradeep Niroula, Oles Shtanko, Alexey V. Gorshkov, Bill Fefferman, and Michael J. Gullans. Tight bounds on the convergence of noisy random circuits to the uniform distribution, 2021.
- [Nap22] John Napp. Quantifying the barren plateau phenomenon for a model of unstructured variational ansätze, 2022.

Response to the Reviewers

We thank the reviewers for their comments on our manuscript, and the generally positive feedback.

Reviewer 1

Reviewer Point P 1.1 — The authors have satisfactorily addressed the comments and the paper is now suitable for publication.

Reply: We thank the reviewer for their kind words.

Reviewer 2

Reviewer Point P 2.1 — I went through the updated manuscript and the response file. The authors have answered four out of five concerns raised by me. However, I am still not convinced regarding point 2.3. In terms of techniques, the work seems a derivative of reference 1 and other existing tools in the literature. I am not sure if Nature Com requires the novelty of the technique as a pre-requisite for the publication. I will be neutral and leave it to the editor to decide. I am okay with acceptance, since most of my concerns have been addressed.

Reply: We thank the reviewer for their overall kind remarks. Following the reviewer's recommendation, we have added more text in the Results section elucidating the novel techniques used in our work.

Reviewer 3

Reviewer Point P 3.1 — The authors addressed all my concerns in a satisfactory manner. I recommend publication.

Reply: We thank the reviewer for their kind words.

Reviewer 4

We thank the reviewer for taking the time to deliberate over our response.

Reviewer Point P 4.1 — The authors clarify the connection between previous results and their work in Corollary 2.2 of the main text (and also in the supplemental material.) The most relevant work is that of [Nap22]. There, it is shown how shallow depth random circuit ansatzes do not exhibit barren plateaus. In the present work, the authors prove that for shallow depth random circuit ansatzes, even though there aren't barren plateaus, the local minimas are far away from the global minima. So, such ansatzes are untrainable because there is a high probability of the optimizer getting stuck in a local minima.

However, although the authors' technical result is novel, the message is not. In [DNS+21, Nap22], it is proven that shallow depth random ansatzes exhibit a severe lack of “anti-concentration.” In other words, the cost function is sharply “peaked” at certain points. Furthermore, [Nap22] proves that the peaks are inverse exponentially suppressed depending on the Hamming weight of the Pauli string under consideration. So, the strings with low Hamming weight (at most $O(\log n)$ Hamming weight) are the only type of strings

that can exhibit sharp speaks. This already makes these types of cost functions mostly useless, because the ansatz is restricted to exploring only an exponentially small fraction of the full space.

So, even though barren plateaus are not exhibited, it was already known that such cost functions are essentially useless for optimization tasks.

Reply: [Nap22] (cited here as [1]) indeed states that low-Hamming-weight Pauli strings are what yield non-barren plateaus, which corresponds to the shallow, local circuit setting we consider here. We also agree that such ansatzes cannot explore the full Hilbert space, and are in that sense restricted (or in other words, are *less expressive*). Still, such ansatzes certainly can express ground states of interest if it is known that the true ground state is in the fraction of Hilbert space explored by the ansatz. As a simple example, a constant depth product state ansatz is sufficiently expressive to find the ground state of 2-local Hamiltonians diagonal in the computational basis, and is known not to exhibit barren plateaus by [1]; however, it is known [2] that this optimization problem generally is NP-hard and exhibits exponentially many poor local minima. In other words, what limits the ansatz’s utility is not *expressivity*, but *untrainability due to poor local minima*. Our average-case results can be seen as an average-case extension of this which includes, for instance, when the Hamiltonian is known to have a near-product ground state that can be prepared with a shallow local circuit. This independence from the expressivity of the ansatz can also be seen in our numerical results, where we have guaranteed the existence of a global minimum in their construction such that there is no concern about the expressivity of the model. We have added clarifying text in the Results section emphasizing this.

Reviewer Point P 4.2 — The authors clarify the strengths of their statistical query model in Sections 2.2 and 2.3 of the paper. Specifically, they clarify that for some regimes, the statistical query model gives worst case hardness results, which is stronger than arguing about average case hardness.

Once again, the technical novelty of the statistical query model and the hardness results the authors prove using that model is interesting. But, I am not sure how novel the messaging is conceptually. It is not clear to me why average case hardness analysis, using loss landscapes, is not already proof enough of the fact that shallow depth ansatzes are bad for optimization tasks. Although a worst case analysis is technically interesting, it does not add too much to the messaging, because such worst-case ansatzes may be inefficient to construct in the first place.

Reply: We included the statistical query analysis as the average-case (i.e., WHRF) analysis leaves open the possibility of trainable ansatzes that exist but are not yet known; it just states that *on average* such ansatzes are untrainable. We emphasize again that the statistical query results we include are not “worst-case” results but are instead, in a sense, “best-case” results—they show that *no* ansatz (in any of the learning settings we consider) is trainable, assuming its training algorithm reduces to statistical queries (which includes, for instance, stochastic gradient descent). We have added more clarifying text in the Results section emphasizing that the statistical query results hold for all ansatzes, and not just in the worst case.

We once again thank the reviewers for their overall kind words, and for generally recommending the article for acceptance.

References

- [1] J. Napp, Quantifying the barren plateau phenomenon for a model of unstructured variational ansätze (2022), arXiv:2203.06174 [quant-ph] .
- [2] L. Bittel and M. Kliesch, Phys. Rev. Lett. **127**, 120502 (2021).